# Genotypic and Phenotypic Characteristics of *Moraxella catarrhalis* from Patients and Healthy Asymptomatic Participants among Preschool Children

**DOI:** 10.3390/pathogens11090984

**Published:** 2022-08-29

**Authors:** Na Zhao, Hongyu Ren, Jianping Deng, Yinju Du, Qun Li, Pu Zhou, Haijian Zhou, Xiangkun Jiang, Tian Qin

**Affiliations:** 1Key Laboratory for Infectious Disease Prevention and Control, National Institute for Communicable Disease Control and Prevention, Chinese Center for Disease Control and Prevention, Beijing 102211, China; 2Zigong Center for Disease Control and Prevention, Control and Prevention of Zigong City, Zigong 643002, China; 3Disease Control and Prevention of Liaocheng City, Liaocheng 252001, China

**Keywords:** *Moraxella catarrhalis*, multilocus sequence typing, preschool children, antimicrobial susceptibility

## Abstract

**(1)****Background**: *M. catarrhalis* can ascend into the middle ear, where it is a prevalent causative agent of otitis media in children, or enter the lower respiratory tract, where it is associated with community-acquired pneumonia (CAP). In this study, we aimed to provide an overview of the prevalence of *M. catarrhalis* in preschool children. **(2)**
**Methods**: *M. catarrhalis* strains were isolated from samples. All isolates were characterized in terms of serotypes (STs), virulence genes, multilocus sequence type, and antibiotic susceptibility. **(3)**
**Results**: The percentages of strains expressing lipooligosaccharides (LOSs), serotype A, B, C, or unknown were 67.61%, 15.71%, 4.28%, and 12.38%, respectively. Among the strains, 185 (88.10%) carried ompB2, 207 (98.57%) carried ompE, and 151 (71.90%) carried ompCD. The most frequently identified STs were ST449 (n = 13), ST64 (n = 11), and ST215 (n = 10). The resistance rates to the antibiotics cefuroxime, azithromycin, and erythromycin were 43.33%, 28.10%, and 39.05%, respectively. **(4)**
**Conclusions**: High prevalence of some-specific ST types and high rates of antibiotic resistance indicate the necessity for an increased vigilance of resistant strains, a rational use of antibiotics in preschool children, and most importantly, the surveillance of healthy asymptomatic participants preschool children with *M. catarrhalis*. Our findings provide a platform for the development of novel *M. catarrhalis* vaccines.

## 1. Introduction

*M. catarrhalis* is a human-restricted, unencapsulated, Gram-negative mucosal pathogen that causes a plethora of diseases, such as acute otitis media, chronic obstructive pulmonary disease, pneumonia, bronchitis, laryngitis, sinusitis, and less frequently, septic arthritis, bacteremia, meningitis, and endocarditis [1,2,3,4]. Moreover, *M. catarrhalis* is recognized as one of the most frequent causes of otorhinolaryngological infections in children. Although the antimicrobial susceptibility of *M. catarrhalis* has remained generally relatively stable, making oral antibiotic (e.g., carbapenems, cephalosporins, fluoroquinolone, and macrolides) treatment feasible for many infections, recent reports have shown an increase in antibiotic-resistant *M. catarrhalis* in China [5,6,7].

*M. catarrhalis* carries certain virulence determinants, such as ompB2, ompE, and ompCD genes and intact LOSs, allowing *M. catarrhalis* to mount a counter-attack to the human immune defense system by resisting complement-mediated bacteriolysis. A prerequisite of the effective prevention and control of *M. catarrhalis* disease is the identification of epidemic clones and the ability to distinguish case clusters or outbreak-related strains from sporadic strains. This can be achieved using the multilocus sequence typing (MLST) of *M. catarrhalis* strains isolated from patients and healthy asymptomatic participants from different geographic regions [2,8,9].

Although there are several reports of the serotype variation or antimicrobial resistance pattern of *M. catarrhalis* in patients [7,10,11], data from molecular subtyping of strains from patients and healthy asymptomatic participants are scarce, especially for preschool children. In the present study, we compared the serotype variation, antimicrobial susceptibility, and molecular epidemiology of *M. catarrhalis* isolates recovered from patients and healthy asymptomatic participants of preschool age to provide an overview of the prevalence of *M. catarrhalis* in this age group.

## 2. Results

### 2.1. Characteristics of M. catarrhalis Isolates

The 210 isolates analyzed in this study were obtained from specimens collected from preschool children with an average age of 2.1 years and a female predominance (57.14%, n = 120). Of the 139 strains collected from patients with an average age of 1.5 years, 50 (35.97%) were isolated from males and 89 (64.03%) from females. Of the 71 strains isolated from healthy preschool children with an average age of 3.7 years, 40 (56.33%) were from males and 31 (43.67%) were from females.

### 2.2. Distribution of LOSs Serotypes and Detection of Virulence Genes

The multi polymerase chain reaction (M-PCR) typing results for the 210 *M. catarrhalis* strains are shown in Table 1. Serotype A (68.74–73.23%) was the most frequent and widely distributed serotype, followed by serotype B (15.71%), non-typeable strains (n = 26; 12.38%), and serotype C (4.28%). There was no significant difference in the variations in serotypes of the strains between those isolated from the patients and those from healthy asymptomatic participants (*p* > 0.05).

Three outer membrane-associated virulence genes encoding outer membrane proteins (OMPs) with different important virulence functions were selected for analysis in the present study. The PCR results from the 210 strains showed that 98.57% were ompE-positive and 88.10% were ompB2-positive. Interestingly, 87.62% of the strains were ompB2-ompE-dual positive, with an almost equal percentage of ompCD-positive and ompCD/ompE-dual positive (71.90 vs. 70.95%), while the ompB2/ompCD-positive and ompB2/ompCD/ompE-triple positive represented the lowest proportions (62.86 vs. 62.38%) (Table 2).

### 2.3. MLST Analysis of 210 M. catarrhalis Isolates

In the analysis of the MLST patterns of the 210 isolates, 105 STs were identified, including 70 new STs, with ST449 (n = 13), ST64 (n = 11), ST215 (n = 10), ST462 (n = 9), and ST394 (n = 8) predominating. Among the 17 different MLST clusters surrounding the STs (Figure 1), six likely primary founders were identified (ST662, ST363, ST224, ST64, STN1, and STN65), corresponding to six CCs (defined as a cluster containing a primary founder): CC662, CC363, CC224, CC64, CCN1, and CCN65. Notably, only 5 of the 17 clusters contained *M. catarrhalis* isolates from only patients or healthy asymptomatic participants, with the remaining 12 clusters containing mixtures of strains from both patients and healthy asymptomatic participants.

Strains from the patients comprised 72 STs, including 46 new STs, with ST449 (n = 12), ST64 (n = 10), ST394 (n = 8), STN1 (n = 7), and ST180 (n = 4) predominating. Those from healthy asymptomatic participants comprised 47 STs, including 29 new STs, with ST215 (n = 6), ST462 (n = 6), ST374 (n = 5), ST363 (n = 4), STN2 (n = 3), and STN9 (n = 3) predominating and no overlap with any predominant ST in the patient group.

### 2.4. Antimicrobial Resistance

Details of the antimicrobial susceptibility of the 210 *M. catarrhalis* strains are listed in Table 3. The highest resistance rate was observed for cefuroxime reaching 43.33% (n = 91), followed by erythromycin (39.05%, n = 82), and azithromycin (28.10%, n = 59). The resistance rates to three other antimicrobials (tetracycline 1.43%; sulfamethoxazole-trimethoprim 0.95%; ciprofloxacin 0.48%) were low (<2.0%). None of the strains showed resistance to imipenem and chloramphenicol. Although not statistically significant, the prevalence of cefuroxime-resistant *M. catarrhalis* was higher in patients (45.32%) than in healthy asymptomatic participants (33.80%, *p* = 0.11). Similarly, the prevalence of azithromycin-resistant *M. catarrhalis* of patients (28.78%) was not significantly different from that of healthy asymptomatic participants (26.76%, *p* = 0.76). In contrast, the prevalence of erythromycin-resistant *M. catarrhalis* from patients (37.41%) was slightly lower than that of healthy asymptomatic participants (42.25%, *p* = 0.45).

Analysis of the susceptibility or resistance of *M. catarrhalis* isolates to azithromycin based on the allelic characteristics of MLST showed that the strains from the patient group comprised 15 clusters and 13 singleton STs, including 10 clusters (66.67%) and three singleton STs (23.07%) that were resistant to azithromycin. Isolates from healthy asymptomatic participants comprised 6 clusters and 21 singleton STs, including 3 clusters (50.00%) and 8 singletons ST (38.10%) that were resistant to azithromycin (Figure 2). Interestingly, in both the patient and healthy groups, ST449, ST64, and ST363 carried multiple resistance to azithromycin, erythromycin and cefuroxime (Figure 2 and Appendix A).

## 3. Discussion

*M. catarrhalis* is a common commensal of the upper respiratory tract. The bacterium can colonize the nasopharynx. Although colonization does not always result in disease, it may be the first step towards invasive disease. Asymptomatic carriers might also become symptomatic patients through bacterial superinfection or after a viral infection or autoimmunity is reduced. References have noted that the carriage rate of *M. catarrhalis* is 25.8–76.6% among global healthy children. In contrast, the detection rate in children patients is 7.6–53.0%. The carriage or detection rate in Europe was 25.8–63.5% vs. 7.6–53.0% (Hungary—63.5 vs. 7.6%; Sweden—52.1 vs. 53.0%; Netherlands—25.8 vs. 10.0–22.0%) [12,13,14,15,16,17]. The carriage or detection rate in Asia was 32.1–76.6% vs. 11.7–30.8% (China—76.6 vs. 13.4%; Korea—35.0 vs. 30.8%; Japan—32.1 vs. 11.7%) [18,19,20,21,22,23]. There was a difference in the detection rate of *M. catarrhalis* between asymptomatic carriers and symptomatic patients. However, for our study, there was no difference in the virulence gene and serotypes distribution between the pathogens obtained by isolation.

The higher virulence of some *M. catarrhalis* strains that cause serious infections is mostly due to the expression of ompE, ompB2, ompCD genes, and intact lipooligosaccharide (LOS). There are already convincing data about their important role in pathogenicity [24,25,26]. The virulence genes ompE and ompCD encode porins that function as adhesion molecules, mediating nutrient transport and serum resistance. OmpB2 determines the resistance to the bactericidal activities of normal serum and its expression significantly increases the virulence of *M. catarrhalis*. In our study, we showed that these three virulence genes were prevalent in isolates recovered from both patients and healthy participants, suggesting that this type of *M. catarrhalis* is carried by healthy preschool children and represents a risk factor for infection. Therefore, attention should be paid to the surveillance of *M. catarrhalis* in healthy preschool children.

As a major virulence factor, LOSs are a main component in the outer membrane of *M. catarrhalis* that are highly conserved among three serotypes [27]. LOSs-based conjugate vaccines derived from the individual serotypes of *M. catarrhalis* are highly immunogenic [28,29,30]. In the current study, we found no significant difference in the distribution of serotypes in strains from patients and healthy asymptomatic participants, suggesting that LOSs-based conjugate vaccines still hold great potential for preventing both symptomatic and asymptomatic *M. catarrhalis* infection.

MLST is widely employed in the epidemiological investigations of outbreaks of various scales as well as sporadic cases. This technique provides high reproducibility and enables inter-laboratory, inter-regional, and inter-national comparison of pathogenic clones. In this study, we used MLST to identify 12 identical clusters (70.59%) of strains in both patients and healthy asymptomatic participants, with ST449 (n = 13), ST64 (n = 11), ST215 (n = 10), ST394 (n = 8), and ST363 (n = 7) as the most common STs. ST449, ST64, and ST363 are three important ST types that are resistant to azithromycin, erythromycin, and cefuroxime. Notably, strains of the STs that overlapped between patients and healthy asymptomatic participants all exhibited antibiotic resistance, in accordance with a previous report that macrolide-resistant *M. catarrhalis* isolates were highly restricted to ST449 and ST363 strains [31].

Unlike other pneumonia-causing bacteria (e.g., *Streptococcus pneumoniae and Haemophilus influenzae*), no vaccine has yet been developed against *M. catarrhalis*, which leaves antibiotic therapy as the only practical and generally empirical means of treating *M. catarrhalis*. Azithromycin is the first-line antimicrobial to treat pneumonia in children [32]; therefore, it is not surprising that azithromycin-resistant strains are emerging with increasing frequency. In contrast to the scarcity of macrolide-resistant *M. catarrhalis* isolates in Europe, North America, and Southeast Asia, their prevalence in China has increased [33,34]. In 2012, 22.5% of *M. catarrhalis* strains in China were reported to be Azithromycin-resistant [35], while the azithromycin resistance rate of the strains in the current study increased to 28.10%. According to our MLST analysis shown in Figure 2, the azithromycin-resistant strains from patients were predominantly distributed in clusters, whereas those from healthy asymptomatic participants were almost all restricted to single STs. Alternatively, all azithromycin-resistant strains come in different regions, thus representing a challenge to monitoring azithromycin-resistant strains in the healthy population. More strikingly, the cefuroxime-resistance rate (4.5%) reported in children (aged < 3 years) with pneumonia during the period 2014–2016 [2] increased sharply to 43.33% in the current study. This alarming increase warrants immediate action in monitoring the prevalence of cefuroxime and azithromycin resistance. Thus, when cephalosporin and macrolide treatment fails in children with *M. catarrhalis* pneumonia, fluoroquinolones should be considered as a better option.

Although we obtained some interesting results, the following limitations of our study should be noted: (i) The isolates recovered for comparison were obtained from specimens from patients and healthy asymptomatic participants who are not from exactly the same geographic locations. However, the distribution of different STs was indeed regionally related, and some important clusters (CC64, CC363, and CC662) and important STs (ST64, ST4363, and ST449) were still detected when comparing different populations. (ii) We cannot completely rule out the possibility that seasonality might have affected the outcome of the study because of the multi-season time span (2016–2018) of the project. In a future study, we will focus on utilizing molecular typing to investigate drug resistance in patients and healthy asymptomatic participants of different ages from the same area. We will name the new alleles and STs, despite the database (http://mlst.warwick.ac.uk/mlst/dbs/Mcatarrhalis/, accessed on 23 July 2022) no longer supporting the uploading of new STs; for this reason, STN1–STN70 have not yet been submitted.

## 4. Materials and Methods

### 4.1. Bacterial Strains

We conducted an observational and retrospective study of 210 *M. catarrhalis* strains collected between 2016 and 2018; 139 strains were isolated from patients and 71 from healthy asymptomatic participants. These strains were isolated from three regions of China. Twenty-three strains were isolated from the sputum of patients with CAP in Sichuan Province (23 strains) and seven from similar patients in Shandong Province, while the remaining 109 strains were isolated from the sputum, bronchoalveolar lavage fluid, conjunctival secretion, and nasal secretions of outpatients with pneumonia, bronchitis, and fever at clinics in Fujian Province. The 71 strains from healthy asymptomatic participants (47 strains from participants in Sichuan Province and 24 from Shandong Province) were isolated from the throat swabs of people who showed no clinical symptoms of infection and had not received antimicrobial therapy during the previous 7 days. These specimens were immediately plated individually on Columbia blood agar plates and incubated for 24 h at 35 °C under 5% CO_2_. The isolates were subsequently subjected to a series of phenotypic identification tests, including the observation of colony morphology and automated identification by mass spectrometry (M-DISCOVER 100, Zhuhai Meihua, Zhuhai, China), according to the manual of clinical microbiology [36]. The strains were stored in milk preservation tubes at −70 °C prior to testing.

### 4.2. DNA Extraction

Chromosomal DNA was extracted from an overnight subculture of each *M. catarrhalis* isolate using TIANamp Bacteria DNA Kit (Tiangen, Beijing China) according to the manufacturer’s instructions.

### 4.3. LOSs Serotyping and Detection of Virulence Genes

Isolates were stereotyped by multiplex polymerase chain reaction (M-PCR) using the primers designed by Verhaegh et al. [37] as shown in Table 4. Genes encoding outer membrane proteins (ompB2, ompCD, and ompE) carried by *M. catarrhalis* isolates were amplified by PCR in a 20 µL reaction mix using the respective primers, which produced PCR products of the expected sizes (shown in Table 4).

### 4.4. MLST

MLST was performed on all 210 isolates by amplifying internal fragments of the following eight housekeeping genes: abcZ (ATP-binding protein), adk (adenylate kinase), efp (elongation factor P), fumC (fumarate hydratase), glyRS (glycyl-tRNA synthetase beta subunit), mutY (adenine glycosylase), ppa (pyrophosphate phospho-hydrolase), and trpE (anthranilate synthase component I) in separate PCR reactions as previously described [2]. The alleles and STs were determined by comparison with the allelic profiles for *M. catarrhalis* shown in the MLST database (http://mlst.warwick.ac.uk/mlst/dbs/Mcatarrhalis/, accessed on 20 July 2022).

### 4.5. Evaluation of Antimicrobial Susceptibility Profiles

Bacterial susceptibility was tested using the broth dilution method for the following antimicrobials: cefuroxime, ciprofloxacin, azithromycin, chloramphenicol, sulfamethoxazole-trimethoprim, imipenem, erythromycin, and tetracycline. Results were evaluated according to protocol M45-A3 (CLSI, 2015) and the standards of the European Committee on Antimicrobial Susceptibility Testing (EUCAST, 2017), using *M. catarrhalis* ATCC 25238 as a reference.

### 4.6. Statistical Analyses

The distribution of *M. catarrhalis* LOSs serotypes, antimicrobial susceptibility rates and levels were subjected to chi-squared or Fisher’s exact tests using SPSS22 software (IBM Corp., Armonk, NY, USA). *p* < 0.05 was set as the threshold for statistical significance. Cluster and minimum spanning trees (MST) were created based on the allelic profiles using BioNumerics software (version 7.1; Applied Maths, at bioMérieux, Craponne, France). In an MLST, a clonal complex (CC) is formed by STs with seven of eight MLST alleles in common with at least three STs, in which the founder ST is defined as the ST with the highest number of single-locus variants (SLVs); single genotypes that did not correspond to any clone groups were defined as singletons. The size of each circle in the MLST indicates the number of strains of that particular type.

## 5. Conclusions

In the current study, we determined the genotypic and phenotypic characteristics of *M**. catarrhalis* from patients and healthy asymptomatic participants among preschool children. The high prevalence of some-specific ST types (ST449, ST363, and ST64) and high rates of antibiotic resistance emphasize the importance of the rational use of antibiotics and the necessity for increased vigilance to detect the occurrence of antibiotic-resistant strains. Importantly, three serotypes were detected in both patients and healthy asymptomatic participants, with no significant difference in the frequencies between the two groups, and some virulence genes were conserved. This information provides a platform for the development of *M. catarrhalis* vaccines in the future.

## Figures and Tables

**Figure 1 pathogens-11-00984-f001:**
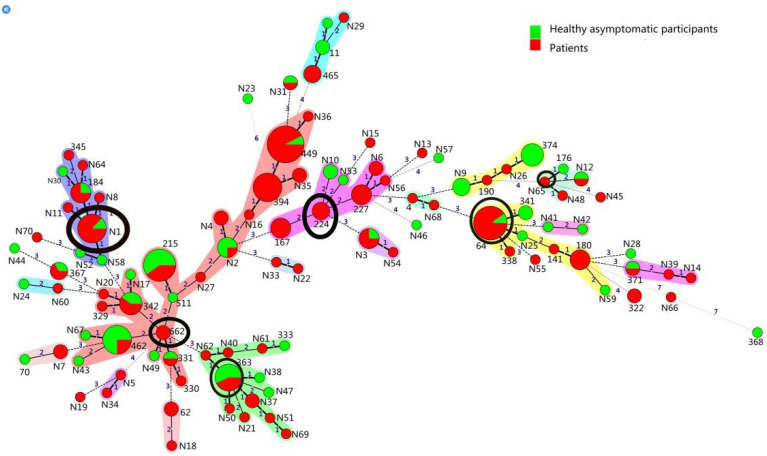
Minimum spanning tree (MST) analysis of 210 *M. catarrhalis* strains based on the allelic profiles generated by MLST. In the MST, the STs are displayed as circles, and the size of a circle indicates the number of strains of this particular ST type. Patients and healthy asymptomatic participants are represented by different colors. The color halo surrounding the STs in Figure denotes STs belonging to different MLST clusters. The likely primary founders with at least three links to other STs are positioned centrally in the cluster and identified by black rings.

**Figure 2 pathogens-11-00984-f002:**
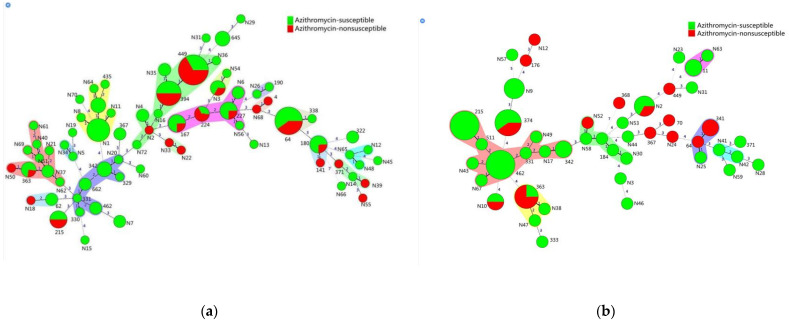
Susceptibility or non-susceptibility of *M. catarrhalis* to azithromycin based on allelic profiles of MLST. (**a**) Different colors indicate the susceptibility or non-susceptibility of patients to azithromycin; and (**b**) different colors indicate the susceptibility or non-susceptibility of healthy asymptomatic participants to azithromycin.

**Table 1 pathogens-11-00984-t001:** Sequence types for lipooligosaccharide serotyping.

Serotypes	Total (N = 210)	Patients (N = 139)	Healthy Asymptomatic Participants (N = 71)
A	142 (67.61%)	90 (64.74%)	52 (73.23%)
B	33 (15.71%)	23 (16.54%)	10 (14.08%)
C	9 (4.28%)	7 (5.03%)	2 (2.81%)
No-serotype	26 (12.38%)	19 (13.66%)	7 (9.86%)

**Table 2 pathogens-11-00984-t002:** Distribution of virulence genes in isolates from patients and healthy carriers.

Virulence Genes	Total (N = 210)		Patients (N = 139)		Healthy Asymptomatic Participants (N = 71)		*p*
	N	%	N	%	N	%	
*ompB2*	185	88.10%	127	91.37%	58	81.69%	0.04
*ompE*	207	98.57%	138	99.28%	69	97.18%	0.23
*ompCD*	151	71.90%	96	69.06%	55	77.46%	0.23
*ompB2* and *ompE*	184	87.62%	126	90.65%	58	81.69%	0.06
*ompB2* and *ompCD*	132	62.86%	88	63.31%	44	61.97%	0.85
*ompE* and *ompCD*	149	70.95%	95	68.34%	54	76.06%	0.24
*ompB2*, *ompE*, and *ompCD*	131	62.38%	87	62.59%	44	61.97%	0.93

**Table 3 pathogens-11-00984-t003:** The non-susceptibility rates of 210 *M. catarrhalis* isolates.

Antimicrobial Agent	Total (N = 210)				Patients (N = 139)				Healthy Asymptomatic Participants (N = 71)			
	S	I	R	Rates of Non- Susceptibility	S	I	R	Rates of Non- Susceptibility	S	I	R	Rates of Non- Susceptibility
CXM	119	66	25	43.33%	76	40	23	45.32%	47	22	2	33.80%
CIP	209	0	1	0.48%	139	0	0	0.00%	70	0	1	1.41%
AZM	151	0	59	28.10%	99	0	40	28.78%	52	0	19	26.76%
C	210	0	0	0	139	0	0	0.00%	71	0	0	0.00%
SXT	208	1	1	0.95%	138	0	1	0.72%	70	1	0	1.41%
IPM	210	0	0	0	139	0	0	0.00%	71	0	0	0.00%
E	128	22	60	39.05%	87	10	42	37.41%	41	12	18	42.25%
TE	207	0	3	1.43%	139	0	0	0.00%	68	0	3	4.23%

CXM, cefuroxime; CIP, ciprofloxacin; AZM, azithromycin; C, chloramphenicol; SXT, sulfamethoxazole-trimethoprim; IMP, imipenem; E, erythromycin; TE, tetracycline.

**Table 4 pathogens-11-00984-t004:** PCR primers used in the present study.

Specific Primer for Gene	Primer Sequence	Product Size (Kb)
*ompCD*-Forward	5′-GTGTGACAGTCAGCCCACTA-3′	1.2
*ompCD*-Reverse	5′-TTGCTACCAGTGATTACTGA-3′	
*ompE*-Forward	5′-TTCAACCCTAACCGCAAC-3′	1.3
*ompE*-Reverse	5′-TTTGGCGTGATAAGCAAG-3′	
*ompB2*-Forward	5′-GCCAGCCTAAGGTTGTCT-3′	2.3
*ompB2*-Reverse	5′-GAAGTTCACGCCAACACG-3′	
LOS 406-Forward (B/C)	5′-CAAAAGAAGACAAACAAGCAGC-3′	C: 4.3;
LOS 408-Reverse (A/B/C)	5′-CATCAAAAACCCCCCTACC-3′	B: 3.3;
LOS 649-Forward (A)	5′-ATCCTGCTCCAACTGACTTTC-3′	A: 1.9

## Data Availability

Not applicable.

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
