# Peer review of "Genotypic and Phenotypic Characteristics of Moraxella catarrhalis from Patients and Healthy Asymptomatic Participants among Preschool Children"

_pathogens, 2022, doi:10.3390/pathogens11090984_

Round 1

Reviewer 1 Report

In this manuscript, Zhao et al. present Moraxella catarrhalis genotypes and phenotypic variants among patients and healthy preschool children. They also report antimicrobial resistance of these isolates. Overall, this work expands the understanding of prevalent M. catarrhalis isolates among this population.

Comments

1. bacterial name italicization not uniform. make sure they are italicized throughout the manuscript.

2. Why was CLSI 2015 protocol used? are there any changes in antimicrobial assay interpretation of M. catarrhalis recently?

Reviewer 2 Report

1.    Table 2. Is there a difference in virulence gene expression between the two groups? The statistics are not provided. 

2.    Does the virulence gene expression differ depending on illness severity?

3.    Figure 1. Yellow circles cannot be seen very clearly. It should be mentioned in the legend that they represent the primary founders.

4.    Line 112. Please provide the exact P value instead of ‘P>0.05’.

5.    Table 3. Is virulence gene expression associated with illness severity, URTI vs. LRTI, or antibiotic resistance?

6.    Lines 160-163: “Based on our results, it can be speculated that some drug-resistant and pathogenic strains of a certain ST cause sickness in people prior to causing asymptomatic infection in healthy preschool children, thus highlighting the importance of close surveillance of some specific STs isolates in the healthy population.” The authors did not provide any time sequence data, or any other data that would support this statement. It seems to be speculation not related to the data presented, and it would be best to delete this statement.

7.    Lines 177-178. What is the distribution of the cefuroxime-resistant strains? Are they from the same region, or are they sporadic?

8.    Line 209 – Correct spelling of “Columbia”

Reviewer 3 Report

A well-written article presenting medically relevant data. Authors present the prevalence and serotypes of Moraxella catarrhalis. Very interesting is that about 30-40% of strains were resistant to the antibiotics cefuroxime, azithromycin, and erythromycin. I would like to propose a few amendments:

1. In the Discussion, it is worth describing what is the symptomatic and asymptomatic M. catarrhalis infection in the world, is it different, for example, in Asia and Europe?

2. Please enter the name and manufacturer of the mass spectrometer.

3. In Institutional Review Board Statement, the approval number of the Ethics Committee should be provided.
